**Flood Risk in a Range of Spatial Perspectives–from Global to Local**
Zbigniew W Kundzewicz[1,2,3], Buda Su[1,4],Yanjun Wang[1], Guojie Wang[1], Guofu Wang[4],
Jinlong Huang[1], Tong Jiang*[1,4]
[1] Collaborative Innovation Center on Forecast and Evaluation of Meteorological
Disasters/Institute for Disaster Risk Management (iDRM), Nanjing University of Information
Science and Technology (NUIST), Nanjing, China
[2]Institute for Agricultural and Forest Environment, Polish Academy of Sciences, Poznan,
Poland
[3]Potsdam Institute for Climate Impact Research, Potsdam, Germany
[4]National Climate Center, China Meteorological Administration, Beijing, China
**Corresponding author:**
Prof. Dr. Tong Jiang
Collaborative Innovation Center on Forecast and Evaluation of Meteorological Disasters/
Institute for Disaster Risk Management (iDRM), Nanjing University of Information Science
and Technology (NUIST), Nanjing, China
**Abstract**
The present paper examines flood risk (composed of hazard, exposure and vulnerability) in a
range of spatial perspectives – from the global to the local scale. It deals with observed records,
noting that flood damage has been increasing. It also tackles projections for the future, related
to flood hazard and flood losses. There are multiple factors driving flood hazard and flood risk
and there is a considerable uncertainty in our assessments, and particularly in projections for
the future. Further, this paper analyses options for flood risk reduction in several spatial
dimensions, from global framework to regional to local scales. It is necessary to continue
examination of the updated records of flood-related indices, trying to search for changes that
influence flood hazard and flood risk in river basins.

**Key words:** flood risk; flood hazard; flood risk reduction; global scale; regional scale; local
scale

## 1. Introduction

River flooding is a major natural disaster, manifesting itself at a range of spatial and temporal scales – from floods on large international rivers conveying huge masses of water (cubic kilometres) lasting over weeks or months to, potentially violent, destructive and killing, inundations in small, often urban, basins, lasting hours. It is estimated that, globally, floods constitute 43% of the total number of natural disasters and 47% of all weather-related disasters, affecting 2.3 billion people in 1995-2015, with the total damage of the order of 662 billion US\$. About 800 million people worldwide are currently living in flood-prone areas and about 70 million of those people are, on average, exposed to floods each year (UNISDR, 2015).

The nature of disastrous floods seems to have changed, in recent decades, with increasing frequency and amplitude of heavy precipitation, flash and urban floods, as well as acute riverine and coastal flooding. The climate track in flood hazard is complex and not ubiquitous (see Section 2). Urbanization and sealing of ground surface have significantly increased surface water runoff in many areas. In some countries, recurrent flooding of crop land has taken a heavy toll in terms of lost agricultural production, food shortages, interrupted food supplies and under-nutrition. However, some deleterious impacts of floods are preventable or at least can be reduced, because of the opportunity of primary prevention through existing, and - in many places – affordable, technologies such as early warning systems and some flood defenses, while awareness raising and education can also be effective in protecting people from adverse impact of floods.

The spatial perspective on floods ranges from a global view by multi-national stakeholders, international organizations, reinsurance institutions, and think-tanks, interested in global affairs to regional (group of countries, river basins which cross national borders, where 40% of global population live and where trans-boundary water issues should be addressed), national, and sub-national (river basins) scales. The local point of view is, for instance, the one of a family of a person who lost life in the flood, of a family that lost their house or workplace in the flood, or of persons responsible for local flood protection. The local scale pertains to the locality and community in flood-prone area, where flood damage incurred and/or where implementation of a flood-risk reduction measure is planned. The global consideration may include aggregation of observation records, model-based projections, as well as international policies aimed at flood risk reduction.

In the present paper, reviewing flood risk in a range of spatial perspectives (from global to local), we start from examination of observed records, noting that flood damage has been

increasing. Further, we discuss projections for the future – flood hazard and flood losses, and
then review flood-risk reduction strategies, starting from the global framework to regional to
local.

**2. Observed records – flood damage has been increasing**
European Academies' Science Advisory Council (see EASAC, 2018), presented the trends in
the number of different types of natural catastrophes worldwide in 1980–2016 (with 1980
levels set at 100%), based on the data from MunichRe NatCatSERVICE. The number of
hydrological events (floods and mass movements) has increased much stronger than the
number of geophysical, meteorological and climatic events. The number of hydrological
events in an average year has now more than quadrupled since 1980 (exceeds 500% in some
years). Global damage caused by "hydrological events", after Munich Re, has been growing,
albeit with strong inter-annual variability (Fig. 1). The named hurricanes, such as the most
costly three that occurred in the North Atlantic in just four weeks: Harvey in August 2017, as
well as Irma and Maria (September 2017) are counted as "meteorological events". However,
the vast majority of the total damage (approximately 95 billion US$) caused by Hurricane
Harvey was related to flooding. This hurricane, that counts as second-costliest on record (after
Katrina), dropped record levels of rain that inundated the city of Houston, Texas, USA. If the
damage caused by flooding related to Harvey were counted in Fig. 1, the year 2017 would
likely be the outstanding one, with highest flood damage ever.

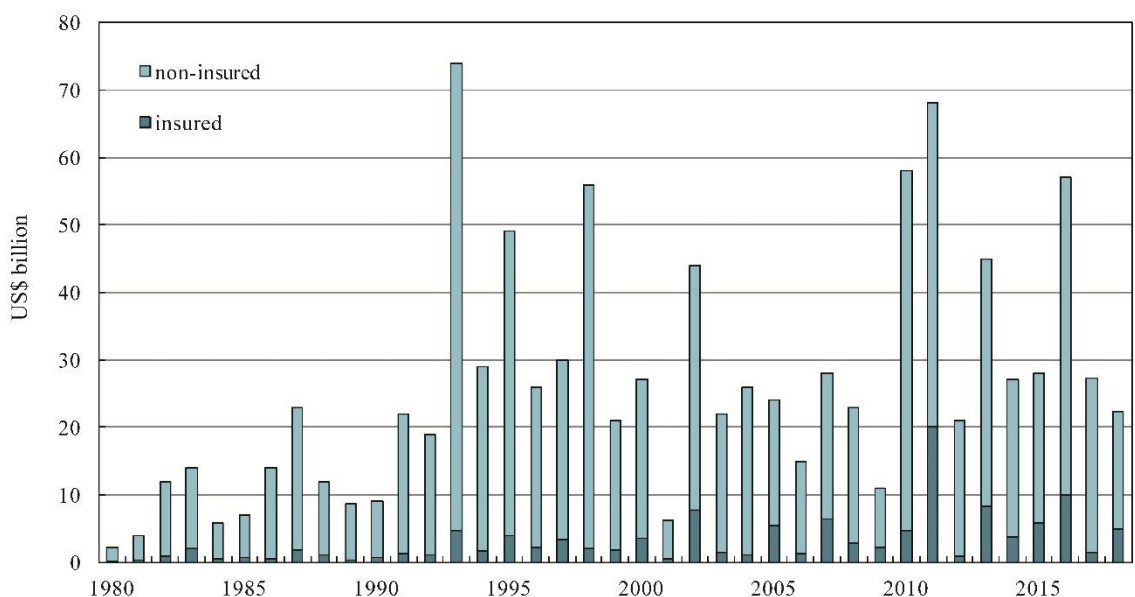


**Fig. 1** Global damage by "hydrological events", in billions US$ (Source: Munich Re NatCatSERVICE).

Flood risk can be assumed to depend on flood hazard, flood exposure and flood vulnerability, which, in turn, are driven by a complex interplay of climate system, terrestrial and hydrological system, as well as the socio-economic system (Fig. 2). Kundzewicz et al. (2014) indicated that increasing exposure of population and assets has been primarily responsible for the recent increase in flood losses.

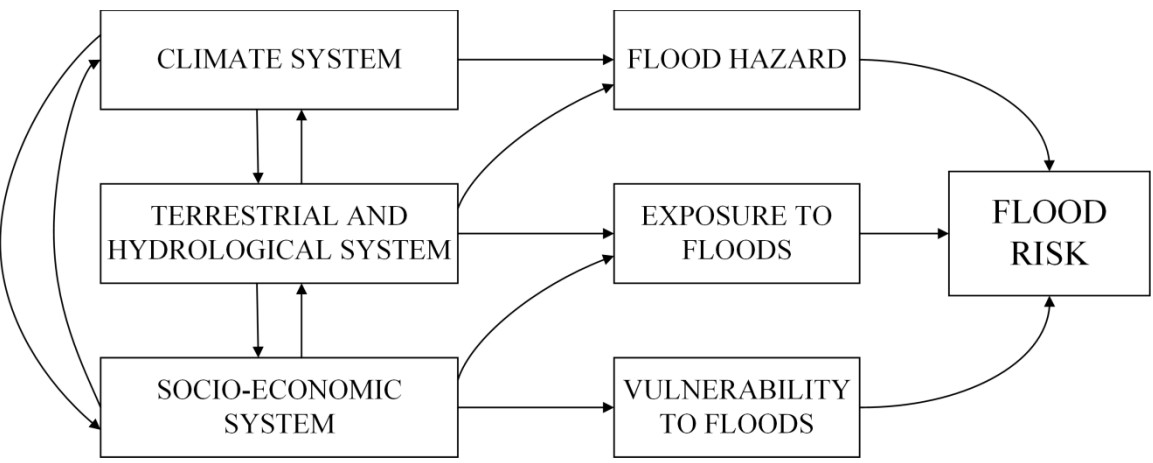

**Fig. 2** Conceptual sketch of components of flood risk and its drivers (after: Kundzewicz et al., 2018c, modified).

Economic losses in monetary units (yet, adjusted for inflation and PPP, i.e. purchase power parity) caused by floods have been on the rise at any spatial scale. They are higher, in absolute terms, in industrialized countries, while relative economic losses expressed as a proportion of GDP and fatality rates are higher in less developed countries. This has grave security implications. This observation holds for natural disasters in general. From 1970 to 2008, over 95% of natural-disaster-related deaths occurred in developing countries (Field et al., 2012).

Typically, disaster losses associated with hydrological extremes can be well buffered in high-income countries (accounting less than 0.1% of GDP), while being much higher, considerably exceeding 1% of GDP in small exposed and less developed countries (Field et al., 2012).

Several factors may explain a perceived increase in flood risk:
- higher frequency and/or intensity of flood events;
- increased exposure of population and assets;

• increase of property value;
• generally, degraded awareness about natural risks, due to less natural lifestyle;
• increased vulnerability; and – not least
• improved and expanded reporting of disasters (sometimes called CNN effect).
We listed vulnerability increase as one of factors that may explain risk increase, but this
holds for some areas only. In general, there is a significant decrease in vulnerability at the global
scale (cf. Kundzewicz et al., 2014; Jongman et al., 2015), largely due to developments in China,
and "vertical urbanization" in particular. Many examples of decreasing vulnerability at the local
scale have been reported (e.g. in Di Baldassarre et al., 2015; Mechler and Bouwer, 2015; Wind
et al., 1999 and Kreibich et al., 2017).
There are countries in the world (see Kundzewicz et al., 2014), where more than 10% of
the population and/or more than 10% of the Gross Domestic Product (GDP) were exposed to
floods in an average year. In absolute terms, the highest number of people exposed was in India
and Bangladesh (over 10 million each), then in China, Vietnam and Cambodia, while the
highest mass of GDP exposed was in the USA and China (over 10 billion US$ per year in each
country), while in India and Bangladesh, it was nearly 10 billion US$. In relative terms, the
highest percentage of people exposed was in Bangladesh and Cambodia (each, over 10% of the
total population), then in Vietnam, while the highest relative share of economy exposed to
floods was estimated in Cambodia and Bangladesh (over 10% in each country), then in
Vietnam.
Dartmouth Floods Observatory (http://floodobservatory.colorado.edu/) has been
compiling information about large floods, worldwide, since 1985. A short list of most deadly
floods (including coastal surges), after the Dartmouth Floods Observatory is presented in Table
1. Among the main causes of the most destructive floods (with more than 1000 fatalities per
event) were: tropical and extra-tropical cyclones, monsoonal rains, tropical storms, torrential
rains, heavy rains, tsunamis, coastal surges, typhoons. Floods with heavy human toll were
recorded in many locations in: Asia (India, China, Bangladesh, Philippines, Afghanistan,
Pakistan, Japan, Myanmar), Central and South Americas (Honduras, Venezuela, Dominican
Republic, Haiti, Salvador, Nicaragua, Costa Rica) and Africa (Tanzania and Sudan).

**Table 1.** Six most deadly floods (including coastal surges), worldwide since 1985.
Information from Dartmouth Floods Observatory

| Countries | Flood beginning | Flood end | Dead [thousand] | Main cause |
|---|---|---|---|---|

| | | | | |
|---|---|---|---|---|
| Thailand | 26.12.2004 | 29.12.2004 | 160 | Coastal surge |
| Bangladesh | 29.04.1991 | 10.05.1991 | 138 | Tropical cyclone |
| Burma | 03.05.2008 | 25.05.2008 | 100 | Tropical cyclone |
| Venezuela, Colombia | 15.12.1999 | 20.12.1999 | 20 | Brief torrential rain |
| Honduras, Panama | 24.10.1998 | 05.11.1998 | 11 | Brief torrential rain |
| India | 29.10.1999 | 12.11.1999 | 9.8 | Tropical cyclone |


150  Frequency and intensity of heavy precipitation have grown in many, but not all, areas of
the globe. However, no gauge-based evidence has been found so far for a clear, widespread,
and consistent change in the magnitude and/or frequency of river floods (see Kundzewicz et
al., 2005; Madsen et al., 2014). Lins and Slack (1999) found that, hydrologically, the
conterminous U.S. had been getting wetter, but less extreme. Later, they (Lins and Slack, 2005)
confirmed the pattern of increasing discharge in the low to moderate range of river flows,
without a concomitant increase in flooding. Relatively few trends in the annual maximum flow
were detected. Hodgkins et al. (2017) examined climate-driven variability in the occurrence of
major floods across North America and Europe, in minimally altered catchments (to eliminate
major non-climatic effects), finding that the number of significant trends was approximately
equal to the number expected due to chance alone. Shaw and Riha (2011) studied three
watersheds in different physiographic regions of New York State, USA and concluded that 20%
or less of annual maximum streamflows were associated with the annual maximum rainfall
events, another 20% - with the annual maximum snowmelt events, while 60% - with moderate
rainfall amounts and very wet soil conditions. Noting that it has not been possible to find
ubiquitous flood hazard changes in observation records in Europe, so far, Kundzewicz et al.
(2018c) detected an increasing trend in the number of large floods, even if the natural variability
is dominating. It is likely that temporally-varying connections exist between indices of climate
variability and variability of the likelihood of destructive abundance of water. Blöschl et al.
(2017) noted no "consistent climate change signal in flood magnitudes" in Europe, while Di
Baldassarre et al. (2010) reported a similar finding for Africa.

171  Blöschl et al. (2017) found climate-induced patterns of change in observed flood timing
in Europe, at the continental scale. They detected earlier spring snowmelt floods throughout NE
Europe (warming-driven change); later winter floods around the North Sea and part of the
Mediterranean coast (related to polar warming) and earlier winter floods in W Europe
(reflecting advancement of soil moisture maxima). In contrast, Lins and Slack (2005) detected
no systematic shift in the timing of the maximum flow in any US region on a monthly time
scale.

## 3. Projections for the future – flood hazard and flood damage


Climate projections show ubiquitous warming for all seasons and most models project increase
in intense precipitation. Seneviratne et al. (2012) presented regional projections of 20-year 24h
precipitation, noting increases over virtually all regions of the Globe.

There have been several global studies of model-based projections of flood hazard, starting

from Milly et al. (2002), who covered selected basins worldwide, and Hirabayashi et al. (2008),
who covered the global scale. It is worthwhile to compare four more recent papers, published
since 2013 by Hirabayashi et al. (2013), Dankers et al. (2014), Arnell and Gosling (2014) and
Giuntoli et al. (2015). Table 2 presents assumptions made in the global projection endeavors
that considerably differ among studies (there are also slightly different reference periods).

**Table 2** Assumptions made in model-based global flood-hazard projection studies.

| Paper | Number of climate model scenarios | Number of hydrological models | Variable of interest | Time horizon of concern | Emission scenario |
|-------|-----------------------------------|-------------------------------|----------------------|-------------------------|-------------------|
| Arnell and Gosling (2014) | 21 GCMs | 1: Mac-PDM.09 | Q100 | 2050s | SRES A1B |
| Dankers *et al.* (2014) | 5 GCMs | 9 GHMs | Q30 | 2070-2099 | RCP8.5 |
| Giuntoli *et al.* (2015) | 5 GCMs | 6 GHMs | Frequency of high flow days | 2066-2099 | RCP8.5 |
| Hirabayashi *et al.* (2013) | 11 GCMs | 1 CaMa-Flood model | Q100 | 2071-2100 | RCP8.5 |


Projections by Hirabayashi et al. (2013) indicate that what used to be a 100-year flood in

the control period in many areas, is likely to occur much more frequently in the future, under
changed climate, with return period of 50 years and below. Hirabayashi et al. (2013) project
increase of hazard (Q100) in most of Asia (except for Western Asia) and in particular –
eastwards of 80ºE. They also project flood hazard to increase in Central Africa in latitude range
20ºS-10ºN and in Central and South America from 20ºN to 40ºS, also in the north of North
America and the East coast of the US. For most of Europe, decrease of flood hazard is projected.
Results of Dankers et al. (2014) referring to a different index, Q30 (30-year 5-day peak flow),
are broadly similar to those reported by Hirabayashi et al. (2013) as to the direction of change,
except for a large area of decrease of hazard in South America. In turn, Giuntoli et al. (2015)
project more frequent days with high river flow conditions over much of the north, from 50ºN
northwards. However, over most of the area of continents, they projected rather small changes,
with absolute value less than 5% (i.e. from -5% to +5%).
Studies of large-scale projections of changes in flood hazard illustrate a considerable
degree of uncertainty. There is no wonder, as projections were determined for different
assumptions (cf. Table 2). They may differ with respect to (see Kundzewicz et al., 2018a,b):
- greenhouse gas emissions scenarios (SRES, RCP);
- driving climate models: general circulation models (GCMs), and regional
climate models (RCMs);
- downscaling techniques and bias correction methods;
- performance of large-scale hydrological models, i.e. global hydrological models
(GHMs) and regional hydrological models (RHMs);
- climate and hydrological model resolution;
- time horizons of future projections;
- reference (historic) intervals;
- return period (recurrence interval) of concern;
- low-temperature effects, e.g. snow and ice component in models;
- general problems related to simulation of extremes and extreme value techniques
applied to time series that are not long enough.
The implications of the changing flood hazard to human society depend on the size of the
population at risk of flooding. Under assumption of a fixed population (at the level of scenario
from 2005), it was projected that annual global flood exposure would increase by about 4±3
times (under RCP2.6), 7±5 times (RCP4.5), 7±6 times (RCP6.0) and 14±10 times (RCP8.5)
from 20th to 21st century (Hirabayashi et al., 2013). However, such results have to be
interpreted with caution, especially considering changing adaptation and risk reduction
capacity.

Where both rain-floods and snow-floods (as well as ice-jam floods) can influence projections, relevant processes and different mechanisms have to be examined, for present and future conditions.

In addition, future flood risk in coastal zones will increase due to the sea level rise (Paprotny and Terefenko, 2017). Taking into account both the socioeconomic pathways and climate change but in absence of further investments in adaptation, Vousdoukas et al. (2018), projected the annual damage caused by coastal flooding in Europe to increase from current 1.25 € billion to 93 – 961 € billion in the end of the 21st century, and the exposed population to increase from the current level of 0.1 million to 1.52 - 3.65 million.

**4. Flood risk reduction – global framework**

Efforts on flood risk reduction are embedded in the general global framework, including the major documents – Hyogo Framework for Action and Sendai Framework for Disaster Risk Reduction.

"Tragedies will continue to be repeated if we do not address water and disaster issues at all levels," stated Dr. Han Seung-soo, the founding chair of the High-Level Experts and Leaders' Panel on Water and Disaster (HELP) ( https://www.unisdr.org/archive/58108), while the UN Special Representative for Disaster Risk Reduction, Ms. Mami Mizutori, remarked that floods which now account for half of all weather-related disasters, highlight how disaster risk reduction is both a long-term development issue and a necessary strategy to prevent disasters and save lives in the short to medium term.

The World Conference on Disaster Reduction held in Hyogo, Japan, in 2005, promoting a strategic and systematic approach to reducing vulnerabilities and risks to hazards, adopted the Framework for Action 2005-2015, identifying ways of building the resilience of nations and communities to disasters (UNISDR, 2007).

Disaster loss has been on the rise with grave adverse consequences for the survival, dignity and livelihood of people, particularly of the poor, and for the hard-won development gains. Disaster risk is increasingly of global concern and a flood occurrence in one region can have an impact on risk in another one (e.g. via broken production links that manifested themselves during and after the 2011 Thailand flood). The Hyogo Framework identified specific gaps and challenges in the following main areas: governance: organizational, legal and policy frameworks; risk identification, assessment, monitoring and early warning; knowledge management and education; reducing underlying risk factors; and preparedness for effective response and recovery.

Disaster risk reduction can be regarded as a cross-cutting issue in the realm of sustainable
development and therefore an important element for the achievement of internationally agreed
Millennium Development Goals.
The global plan for reducing disaster losses, the Sendai Framework for Disaster Risk
Reduction, 2015-2030, was adopted by UN Member States in 2015, at the Third UN World
Conference on Disaster Risk Reduction in Sendai, Japan
(https://www.unisdr.org/we/coordinate/sendai-framework). It is a voluntary, non-binding,
agreement aimed at a substantial reduction of disaster risk and losses in lives, livelihoods and
health and in the assets. It emphasizes the importance of risk-informed investment in critical
infrastructure, including water facilities, to avoid the creation of new risk. Disaster risk
reduction and prevention should be integrated in long-term national planning and education on
disaster risk must be advanced. Recognizing the State's primary role to reduce disaster risk but
also noting that responsibility should be shared with stakeholders, the Sendai Framework
agreement, aiming to make a difference for poverty, health and resilience is the major document
of the recent development agenda, embracing seven targets and four priorities for action.
The global targets include substantial reduction of mortality in flood disasters and the
number of affected people, reduction of direct economic loss and damage to critical
infrastructure as well as disruption of basic services (among them health and educational
facilities), including through enhancing resilience (recovery). They also include work on
national and local disaster risk reduction strategies, on international cooperation and on
increasing the availability of and access to early warning systems (also dedicated to multiple
hazards) and disaster risk information and assessments. Timelines for achieving these targets
and reference intervals for measuring the progress were defined.
The priorities for action refer to understanding of disaster risk in its dimensions of
vulnerability, capacity, exposure of persons and assets, hazard characteristics and the
environment. Such knowledge can be used for risk assessment, as well as to various flood risk
reduction strategies - prevention, mitigation, preparedness and response, recovery and
rehabilitation (see Dieperink et al., 2016, Driessen et al., 2016, Hegger et al., 2016 and
Kundzewicz et al., 2018b). Strengthening disaster risk governance at a range of levels (national,
regional and global) is another priority. Also investing in disaster risk reduction to enhance the
economic, social, health and cultural resilience of persons, communities, countries and their
assets, as well as the environment is an identified priority. So is also enhancing disaster
preparedness for effective response and "Building Back Better". Disaster risk reduction has to
be integrated into sustainable development measures.
Willner et al. (2018) computed the increase in flood protection that would be required
worldwide for subnational administrative units, in order to keep the historic high-end fluvial
flood risk in the next 25 years. They found that most of the United States, Central Europe, and
Northeast and West Africa, as well as large parts of India and Indonesia, require strong
adaptation effort. For example, according to the results of this paper, flood protection needs to
at least double over more than half of the United States, within the next two decades.
However, the increase of flood protection levels to meet the requirements posed by Willner
et al. (2018) would lead to having even more levees, that attract even more people and assets in
flood-prone areas (that are often assumed to be perfectly safe by inhabitants). Since the seminal
work of Gilbert White in the 1940s (White, 1945), many authors reported on safe-development
paradox, residual risk and adverse levee effects (e.g. Kates et al., 2006; Ludy and Kondolf,
2012; Di Baldassarre et al., 2014). It has been shown that the introduction or reinforcement of
structural protection measures are often associated with negative effects. Such effects include
increasing exposure to flooding (Kates et al., 2006) and increasing vulnerability to flooding (as
protected flood-prone areas are perceived as safer, so that inhabitants have less incentives to
take individual precautionary measures; see Ludy and Kondolf, 2012). There is a social
injustice effect - structural flood protection measures may alter the spatial distribution of risk
in a way that affects less privileged social groups (Di Baldassarre et al., 2014). People in
structurally protected areas are less willing to relocate from risky areas (Mård et al., 2018).
Furthermore, levees that prevent natural inundation of floodplains also adversely affect
biodiversity and ecological functions (Auerswald et al., 2019), e.g. via elimination of a "flood
pulse".

**5. Flood risk reduction – from regional to local**
There is no doubt that flood risk has grown in many places and is likely to grow further in the
future, due to a combination of anthropogenic and climatic factors. Intense precipitation grows
in the warming climate. However, reliable and detailed quantification of aggregate flood
statistics is very difficult to obtain for the past-to-present and is virtually impossible to obtain
for the future. Nevertheless, despite of the lack of reliable projections, flood risk reduction
endeavors have been carried out at a range of scales, from regional (multi-national) to national,
sub-national and local.
At the sub-continental scale, European Union (EU) passed a dedicated Directive
2007/60/EC on the assessment and management of flood risks (EU 2007), that required all EU
Member States (28 at present) to identify areas at risk from flooding, to map the flood extent as

well as assets and humans at risk in these areas and to take adequate and coordinated measures to reduce this flood risk. This Directive also reinforces the rights of the public to access information and to participate in the planning process. The Directive aims to reduce and manage the risks that floods pose to human health, economic activity, environment, and cultural heritage. The Directive required EU Member States to establish flood risk management plans focused on prevention, protection and preparedness by 2015.

Presence of people and wealth in flood prone areas can be regarded as an illness. One can prevent the risk, by keeping the destructive water away from people and proceeding with flood defenses. This is the curation of the symptoms of the illness. One can also keep people away from the destructive water by way of zoning and ban on floodplain development. This is curation of the source of the illness. But, it is also necessary to prepare to living with floods. This embraces flood mitigation – keeping water where it falls, flood preparation – forecasting, warning, as well as preparation for evacuation and the post-flood recovery (see Dieperink et al., 2016; Driessen et al., 2016; Hegger et al., 2016; Nieland and Mushtaq, 2016, Kundzewicz et al., 2018).

Since it is naïve to expect availability of trustworthy quantitative projections of future flood hazard (as some practitioners clearly do), in order to reduce flood risk, one should focus attention on identification of existing risk and vulnerability hotspots and improve the situation in areas where such hotspots occur (Kundzewicz et al., 2017).

The prerequisite for flood risk reduction is to examine long time series of reliable records on flood-related information. Koç and Thieken (2018) carried out a comparative national review of information on floods in Turkey from three sources: Turkey Disaster Database (TABB), the Emergency Events Database (EM-DAT), and the Global Active Archive of Large Flood Events—Dartmouth Flood Observatory. They found large mismatches in the flood data for Turkey, related to the number of events, the number of affected people and the economic loss.

Flood protection, i.e. adaptation to huge variability of discharge, has been developed in China for four millennia, since the quasi-legendary Emperor Yu, who established the Xia dynasty, marking the beginning of Chinese civilization. He succeeded in taming a long-lasting and disastrous flood in the Yellow River basin by dredging and channelling the rivers to drain the floodwaters and

Flood protection has always been important in China, where hundreds of millions of people live in river valleys. Structural measures, both dikes and dams of different sizes, have a very long tradition in China (a term "hydraulic civilization" was coined by Wittfogel, 1956)

and continue to play a vital role in flood prevention also today, and in the foreseeable future.
The multi-objective, massive Three Gorges Dam on the River Yangtze, the world's greatest
engineering work, has flood protection as the principal objective. Many large reservoirs, also
with flood protection as the main objective, have been built in China, with a total storage
capacity in excess of $0.5 \times 10^{12}$ m$^3$, accounting for over one fifth of the total estimated annual
runoff from the land areas (Guo et al., 2004). Typically, water storage reservoirs serve multiple
purposes: flood control, hydropower, irrigation, water supply, navigation, etc. The total number
of large dams has increased very strongly since 1960, when only five large dams (higher than
100 m) existed in China. The number of large dams grew tenfold in 2000 (Xu et al., 2010). In
the second half of the 20th century, more than 200 thousand kilometers of dikes have been
strengthened for alleviating the impacts of floods in China (Zhang et al., 2002).
The level of expenditure on flood protection in China has grown considerably in recent
decades. However, despite the massive efforts, it is getting abundantly clear that complete flood
control is not possible. Even if there exist powerful levees along the rivers in China, they may
not provide satisfactory protection of the riparians during large floods (see Kundzewicz and
Xia, 2004). Increasingly, large flood damage has been recently occurring on medium- and
small-size rivers. Hence, improvement of flood risk management is needed in the country and
ambitious and vigorous attempts to improve flood preparedness have been already undertaken,
by both structural ("hard") and non-structural ("soft") measures. The former refer to such
defences as dikes, dams and flood control reservoirs, diversions, etc. The latter include
implementing watershed management (source control), zoning; insurance; flood forecasting–
warning system; and awareness raising (Surminski et al., 2015; Nieland and Mushtaq, 2016;
Adelekan and Asiyanbi, 2016). The coping capacities at a local level can influence the
robustness of flood warning system (Daupras et al., 2015).
In many countries, flood protection is distributed among several agencies, hence effective
cooperation and communication among federal, state and local stakeholders is essential. This
is inherently difficult, but progress has been achieved in China in flood forecasting integration,
data sharing and collaborative problem solving. The China Meteorological Administration
(CMA) collects observations of precipitation and other meteorological variables and prepares
precipitation forecasts. The Ministry of Water Resources (MWR) of China collects
hydrological observations (e.g., of river levels and discharges) and is responsible for flood
forecasting and dissemination of the forecast. River basin commissions in China (altogether –
seven commissions, including the Yangtze River Basin Commission) are agencies of the MWR.
The Flood Prevention Law of 2007 laid out principles and responsibilities for flood prevention

planning in China. There is a national standard (GB50201-94) drafted by the Ministry of Water Resources and issued by the Ministry of Construction in 1994 dealing with flood return periods for different categories of location (Gemmer et al., 2011). In 2010, flood hazard mapping guidelines were published as a professional standard by the Ministry of Water Resources.

Gemmer et al. (2011) reviewed climate change adaptation in China, the National Climate Change Programme and China´s White Paper "China's Policies and Actions for Addressing Climate Change". All 34 provinces of China produced a climate change adaptation plan, including flood risk reduction.

It is a well established observation that occurrence of a disastrous flood event in a country or a region improves awareness and triggers investment in flood risk reduction as well as funding of relevant research. In fact, there are many case studies that report social learning effects, one of the findings being that the negative impact of an extreme flood tends to be lower if such an event occurs shortly after another one (e.g. in Jongman et al., 2015; Di Baldassarre et al., 2015; Mechler and Bouwer, 2015; Wind et al., 1999 and Kreibich et al., 2017). Di Baldassarre et al. (2015) show adaptation effects in study areas around the world, while Mechler and Bouwer (2015) noted decreasing number of flood fatalities in Bangladesh over the past decades. Wind et al. (1999) reported that the economic losses of the 1995 Meuse River flooding were much lower than those in 1993, even though the magnitudes of the two events were comparable. Kreibich et al. (2017) illustrated the learning dynamics by way of multi-regional, paired, flood event studies. However, sometimes deficiencies in learning show up. Marks and Thomalla (2017) studied consequences of the great 2011 flood in Thailand, noting that only minor efforts to reduce flood risk were made. The socio-political transformation needed to reduce system vulnerability has not occurred. The focus was on structural defenses - building floodwalls to reduce risk to large-scale enterprises, and this results in redistribution of risk to unprotected areas.

## 6. Concluding remarks

Many studies of flood hazard projections demonstrate the likely rise of flood hazard in the future. Plausible climate change scenarios indicate the possibility of increases in both the frequency and the magnitude of flooding events in many regions. Yet there has been no conclusive and general finding as to how climate change affects flood behavior, in the light of data observed so far, except of some indications of regional changes in timing of floods observed in some areas, with increasing late autumn and winter floods (caused by rain) and less

ice-jam-related floods, e. g., in Europe. The natural variability in observation records is
overwhelming.

The flood risk depends on a combination of anthropogenic and natural factors, such as

climate, land use, as well as population density and wealth (hence – damage potential) in flood-
risk areas and development of flood defenses. Owing to the growing population pressure,
activities like deforestation, agricultural land expansion, urbanization (and increasing sealing
of the ground surface), construction of roads, as well as reclamation of wetlands and lakes have
been progressing. This has reduced the available water storage capacity in river basins,
increased the value of the runoff coefficient, and aggravated flood hazard and flood risk. Flood
potential has ubiquitously increased – there is simply more to lose.

There are multiple factors driving flood hazard and flood risk and there is a considerable

uncertainty in our assessments, and in particular projections for the future. In many places flood
risk is likely to grow, due to a combination of anthropogenic and climatic factors. However, in
general, it is difficult to disentangle the climatic change component in maximum river flow or
flood hazard records from strong natural variability and direct, man-made, environmental
changes. There is a large difference between flood hazard projections obtained by using
different scenarios and different models. Therefore, one should be careful with flat-rate
statements on changes in flood hazard and flood risk, and on climate change impact in
particular. The impact of climate forcing on flood risk is complex and depends on the flood
generation mechanism. Indeed, higher and more intense precipitation has been already observed
in many (but not all) areas of the Globe and this trend is expected to strengthen in the warmer
world, directly impacting on flood risk. Therefore, common-sense changes to design rules,
aimed at flood risk reduction, have been introduced in some countries of Europe, based more
on precautionary principle rather than on robust science. The design flood was adjusted upward
in light of projections for the warmer climate.

However, it is a robust statement that, in general, today's climate models are still not good

enough at producing local climate extremes due to, *inter alia*, inadequate (coarse) resolution.
There is hope that, with improving resolution, models will be able to grasp details of extreme
events in a more accurate and reliable way (Kundzewicz and Schellnhuber, 2004).

It is necessary to continue examination of the updated records of flood-related indices,

trying to search for changes that influence flood hazard and flood risk in river basins. Possibly,
there have been and will continue to be changes in intense precipitation; changes in cyclone
track; changes in land use; and changes in exposure and vulnerability. Early detection and
attribution of changes at any spatial scale would be of vast practical importance.

**Acknowledgments**

This study was supported by National Key Research and Development Program of China MOST (2018FY10050001) and bilateral cooperation project between the Natural Science Foundation of China and the Pakistan Science Foundation (41661144027). The authors are thankful for the support by the High-level Talent Recruitment Program of the Nanjing University of Information Science and Technology (NUIST). Thanks are also due to Munich Re NatCatSERVICE for provision of global flood loss data. The review by two anonymous referees, who provided many constructive and useful comments proved to be valuable and allowed us to enrich this paper.

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
