# Peer review of "1. Introduction"

_Natural Hazards and Earth System Sciences, 2018_

## Referee Comment (RC1) · Anonymous Referee #1 · 16 Feb 2019

This paper discussed a very important topic, the flood risk at global and regional scales. I appreciate the authors' effort to tackle this issue. But it requires a decent amount of work to solidify some statements.

I suggest the authors extend and solidify the literature review in this paper. For example, the authors believe that the frequency of flood does not have obvious changes during the past. However, based on previous research, at least in the US, the low-frequency floods do have a significant change, while the high-frequency floods do not have a significant change.

The characteristics of floods include are magnitude, frequency and timing. So I also suggest the authors extend a paragraph to discuss the influence of the change of floods' timing, instead of just one sentence in the current version.

[Figure]

Please change the format of latitude and longitude.

---

## Referee Comment (RC2) · Anonymous Referee #2 · 1 Mar 2019

I have really enjoyed reviewing this manuscript, which examines flood risk from global to local perspectives. The topic is very relevant, and such a paper is timely. Yet, I think that there are four key aspects that have been neglected (or not sufficiently well discussed). This manuscript would highly benefit from including a critical discussion around the following four major points (A-D).

A) Line 44-53.

This paragraph requires references, as some of these statements are in fact contested by many scholars. Flood trends reported so far are not so strong. For example, in their Science paper, Blöschl et al. (1) states that: "Will a warming climate affect river floods? The prevailing sentiment is yes, but a consistent signal in flood magnitudes has not been found." "A warming climate is expected to have an impact on the magnitude

and timing of river floods; however, no consistent large-scale climate change signal in observed flood magnitudes has been identified so far." "Existing studies have been unable to identify a consistent climate change signal in flood magnitudes." Blöschl et al. (1) mainly refer to floods in Europe, but similar outcomes where found in other places around the world, such as Africa (2).

B) Line 111.

"Increased vulnerability" is listed as a factor for increasing flood risk. It is important to say that in fact at the vulnerability is in fact decreasing at the global scale, as shown for example on the PNAS paper by Jongman et al. (3). At the local scale, there are indeed instances in which vulnerability is increasing, but many authors have shown several examples of decreasing vulnerability (e.g. 4-7 among many others). I am aware that good news and promising trends sell less than bad news and catastrophic trends, but I think these outcomes should still be recognized in a scientific paper. See also my point D below.

C) Lines 263-268 (and following Section 5).

Previous sections have discussed that flood risk is increasing because more and more people live in flood-prone areas. This is a globally accepted fact. However, in this section the authors suggest increasing protection levels and having even more dykes or levees, which have been shown to attract even more people in flood-prone areas!! There is more than abundant literature on safe-development paradox, residual risk and levee effects (e.g. 8-11 to cite only a few) since the work of Gilbert White in the 1940s (8). Numerous scholars have showed that the introduction or reinforcement of structural protection measures are often associated with negative effects, such as: Increasing exposure to flooding. As protected flood-prone areas are perceived as safer, they attract more assets and people (9). Increasing vulnerability to flooding. As protected flood-prone areas are perceived as safer, people living in these areas have less incentives to take individual precautionary measures (10) Social injustice. Structural

measures protecting same areas from frequent flooding, alter the spatial distribution of risk in a way that can affect social groups that are less privileged (11). Preventing relocation. People is highly protected areas are less willing to relocate from risky areas (12). Losses of biodiversity. Levees and dikes that prevent the natural inundation of floodplain also negatively affect biodiversity and ecological functions (13).

D) Lines 352-358.

This paragraph, which deals with social learning, is too shallow. "It is assumed. . ." not clear by whom, and in which context. There is abundant literature in this topic, which deserves a better treatment. Instead, a specific example is provided (2011 flood in Thailand) to hint that such a learning is not really happening. In fact, there are many case studies showing learning effects or that the negative impact of an extreme event tends to be lower if such an event occurs shortly after a similar one (e.g. 3-7 among many others): Decreasing flood fatalities have been observed in Bangladesh over the past decades (4). The economic losses of the 1995 Meuse River flooding in Central Europe were remarkably lower than those in 1993, even though the magnitudes of the two events were similar (5). Di Baldassarre et al. (6) show adaptation effects in study areas around the world. Kreibich et al. (7) show multiple examples of learning dynamics in several test sites. Vulnerability to river flooding has been declining over the past decades (3), as a result of adapting response at the local scale.

REFERENCES

1. Blöschl, G., Hall, J., Parajka, J., Perdigão, R. A., Merz, B., Arheimer, B., et al. (2017). Changing climate shifts timing of European floods. Science, 357(6351), 588-590.

2. Di Baldassarre, G., A. Montanari, H. Lins, D. Koutsoyiannis, L. Brandimarte, and G. Bloeschl (2010). Flood fatalities in Africa: from diagnosis to mitigation, Geophysical Research Letters, 37, L22402.

3. Jongman, B., Winsemius, H. C., Aerts, J. C., de Perez, E. C., van Aalst, M. K.,

Kron, W., & Ward, P. J. (2015). Declining vulnerability to river floods and the global benefits of adaptation. Proceedings of the National Academy of Sciences, 112(18), E2271-E2280.

4. Mechler, R., & Bouwer, L. M. (2015). Understanding trends and projections of disaster losses and climate change: is vulnerability the missing link? Climatic Change, 133(1), 23-35. 5. Wind, H. G., Nierop, T. M., de Blois, C. J., & de Kok, J. L. (1999). Analysis of flood damages from the 1993 and 1995 Meuse floods. Water Resources Research, 35(11), 3459-3465.

6. Di Baldassarre, G., Viglione, A., Carr, G., Kuil, L., Yan, K., Brandimarte, L., & Blöschl, G. (2015), Perspectives on socio-hydrology: Capturing feedbacks between physical and social processes, Water Resources Research, 51, 4770–4781.

7. Kreibich, H., Di Baldassarre, G., Vorogushyn, S., Aerts, J. C., Apel, H., et a. (2017). Adaptation to flood risk: Results of international paired flood event studies. Earth's Future, 5(10), 953-965.

8. White, G.F. (1945). Human Adjustments to Floods. Department of Geography Research Paper No. 29. Department of Geography, University of Chicago, Chicago. 225 pages.

9. Kates, R. W., Colten, C. E., Laska, S., and Leatherman, S. P. (2006), Reconstruction of New Orleans after Hurricane Katrina: A research perspective. Proceedings of the National Academy of Sciences of USA, 103(40), 14653-14660.

10. Ludy J., & Kondolf, G. M. (2012). Flood risk perception in lands "protected" by 100-year levees. Natural Hazards, 61(2), 829-842.

11. Di Baldassarre, G., Kemerink, J.S., Kooy, M., Brandimarte, L. (2014). Floods and societies: the spatial distribution of water-related disaster risk and its dynamics. Wiley Interdisciplinary Reviews: Water, 1(2), 133-139.

12. Mård, J., Di Baldassarre, G., Mazzoleni, M. (2018) Nighttime light data reveal how

flood protection shapes human proximity to rivers. Science Advances, 4(8), eaar5779.

13. Auerswald, K., Moyle, P., Seibert, S. P., & Geist, J. (2019). HESS Opinions: Socio-economic and ecological trade-offs of flood management–benefits of a transdisciplinary approach. Hydrology and Earth System Sciences, 23(2), 1035-1044.

---

## Author Comment (AC1) · 1 Mar 2019

Authors' response to Interactive comment on "Flood Risk in a Range of Spatial Perspectives – from Global to Local" by Z. W. Kundzewicz et al.

Anonymous Referee #1

This paper discussed a very important topic, the flood risk at global and regional scales. I appreciate the authors' effort to tackle this issue. But it requires a decent amount of work to solidify some statements.

Thanks for encouraging words. We tried to solidify questioned statements.

I suggest the authors extend and solidify the literature review in this paper. For ex-

ample, the authors believe that the frequency of flood does not have obvious changes during the past. However, based on previous research, at least in the US, the lowfrequency floods do have a significant change, while the high-frequency floods do not have a significant change.

We did our best to extend the literature review, even if there are thousands of source items that could be referred to, so that we had to be selective. Indeed, we introduced additional US references (Links and Slack, 1999, 2005; as well as Shaw & Riha, 2011) that solidify the review and demonstrate the subtleties of findings, demonstrating the lack of a one-size-fits-all result.

The characteristics of floods include are magnitude, frequency and timing. So I also suggest the authors extend a paragraph to discuss the influence of the change of floods' timing, instead of just one sentence in the current version. We devoted more room to the timing issues, referring to results by Blöschl et al. (2017) as well as Hodgkins et al. (2017). Please change the format of latitude and longitude. We changed the format by using superscripts.

Please also note the supplement to this comment:
https://www.nat-hazards-earth-syst-sci-discuss.net/nhess-2018-336/nhess-2018-336-AC1-supplement.pdf

---

## Author Comment (AC2) · 4 Mar 2019

I have really enjoyed reviewing this manuscript, which examines flood risk from global to local perspectives. The topic is very relevant, and such a paper is timely. Yet, I think that there are four key aspects that have been neglected (or not sufficiently well discussed). This manuscript would highly benefit from including a critical discussion around the following four major points (A-D).

Many thanks for these kind words and, above all, for excellent, constructive review that helped us enrich our paper. We reacted to all points raised by this referee.

A) Lines 44-53 of the original manuscript / lines 44-54 of the revised manuscript This

paragraph requires references, as some of these statements are in fact contested by many scholars. Flood trends reported so far are not so strong. For example, in their Science paper, Blöschl et al. (1) states that: "Will a warming climate affect river floods? The prevailing sentiment is yes, but a consistent signal in flood magnitudes has not been found." "A warming climate is expected to have an impact on the magnitude and timing of river floods; however, no consistent large-scale climate change signal in observed flood magnitudes has been identified so far." "Existing studies have been unable to identify a consistent climate change signal in flood magnitudes." Blöschl et al. (1) mainly refer to floods in Europe, but similar outcomes where found in other places around the world, such as Africa (2).

Indeed, we are well aware that the climate track in flood hazard is generally weak and by no means it is ubiquituous. We carefully selected the wording used in our paper, in order not to create impression that we see a major climatic pattern in flood hazard. We added one sentence in lines 46-47 but we discuss the climate track in more detail in Section 2 (lines 150-170). We refer to the important work by Blöschl et al. (2017), mainly conveying their principal finding – detection of changes in pattern of flood timing. Indeed, Blöschl's publication in Science is perhaps the highest place where one can find a statement that "a consistent large-scale climate change signal in observed flood magnitudes has not been identified so far". (Attention: there was also a paper: Mudelsee, M., Börngen, M., Tetzlaff, G. & Grünewald, U. (2003) No upward trends in the occurrence of extreme floods in central Europe, Nature, 421, 166-169). However, there have been many other works published before Blöschl et al. (2017), reporting a similar finding (e.g. Lins & Slack, 1999, 2005; the book edited by Kundzewicz, 2012 and cited in Blöschl et al., 2017; as well as Madsen et al., 2014). We could identify some (not persuading) changes in selected indices (Kundzewicz et al., 2005, 2018c). We incorporated both references proposed by referee #2 in part (A) of the review.

B) Line 111 of the original manuscript / line 119 of the revised manuscript

"Increased vulnerability" is listed as a factor for increasing flood risk. It is important to say that in fact at the vulnerability is in fact decreasing at the global scale, as shown for example on the PNAS paper by Jongman et al. (3). At the local scale, there are indeed instances in which vulnerability is increasing, but many authors have shown several examples of decreasing vulnerability (e.g. 4-7 among many others). I am aware that good news and promising trends sell less than bad news and catastrophic trends, but I think these outcomes should still be recognized in a scientific paper. See also my point D below.

Very good point. We added a paragraph (lines 121-126), presenting caveats and explanations, as proposed by referee #2. We incorporated all references proposed in part (B) of the review.

C) Lines 263-268 of the original manuscript / lines 303-318 of the revised manuscript

Previous sections have discussed that flood risk is increasing because more and more people live in flood-prone areas. This is a globally accepted fact. However, in this section the authors suggest increasing protection levels and having even more dykes or levees, which have been shown to attract even more people in flood-prone areas!! There is more than abundant literature on safe-development paradox, residual risk and levee effects (e.g. 8-11 to cite only a few) since the work of Gilbert White in the 1940s (8). Numerous scholars have showed that the introduction or reinforcement of structural protection measures are often associated with negative effects, such as: Increasing exposure to flooding. As protected flood-prone areas are perceived as safer, they attract more assets and people (9). Increasing vulnerability to flooding. As protected flood-prone areas are perceived as safer, people living in these areas have less incentives to take individual precautionary measures (10) Social injustice. Structural measures protecting same areas from frequent flooding, alter the spatial distribution of risk in a way that can affect social groups that are less privileged (11). Preventing relocation. People is highly protected areas are less willing to relocate from risky areas (12). Losses of biodiversity. Levees and dikes that prevent the natural inundation of floodplain also negatively affect biodiversity and ecological functions (13).

Thanks for indicating the areas that require strengthening. We touched upon many points raised by reviewer #2 in part C, i.e. we did not neglect them in the first place. However, indeed we did not discuss them sufficiently well. It is quite a rare case that a reviewer sacrifices a lot of time to produce a set of specific, constructive, remarks with which co-authors agree with delight, so that a win-win situation arises. We introduced a new paragraph in lines 303-318 that discusses the points proposed by reviewer #2 in part C of the review. We also cited all references proposed by reviewer #2 in part C of the review.

D) Lines 352-358 of the original manuscript / lines 407-423 of the revised manuscript

This paragraph, which deals with social learning, is too shallow. "It is assumed: : :" not clear by whom, and in which context. There is abundant literature in this topic, which deserves a better treatment. Instead, a specific example is provided (2011 flood in Thailand) to hint that such a learning is not really happening. In fact, there are many case studies showing learning effects or that the negative impact of an extreme event tends to be lower if such an event occurs shortly after a similar one (e.g. 3-7 among many others): Decreasing flood fatalities have been observed in Bangladesh over the past decades (4). The economic losses of the 1995 Meuse River flooding in Central Europe were remarkably lower than those in 1993, even though the magnitudes of the two events were similar (5). Di Baldassarre et al. (6) show adaptation effects in study areas around the world. Kreibich et al. (7) show multiple examples of learning dynamics in several test sites. Vulnerability to river flooding has been declining over the past decades (3), as a result of adapting response at the local scale.

Again, a very useful comment. Indeed, we did not dare to use the term "social learning", therefore stating that "[t]his paragraph, which deals with social learning, is too shallow" is an euphemism on the part of referee #2. However, adopting the recommendations of referee #2 we tried to render this paragraph (lines 407-423) a little less "shallow".

Nevertheless, we are aware that there is a vast body of literature on the topic that we do not cite and many aspects that deserve to be tackled and are ignored in this (already quite comprehensive) paper. Prioritization was needed when selecting material for the general review undertaken in our paper. We think that an attempt to offer an objective prioritization would be a mission impossible. We cited all references proposed by this reviewer in part (D) of the review.

REFERENCES 1. Blöschl, G., Hall, J., Parajka, J., Perdigão, R. A., Merz, B., Arheimer, B., et al. (2017). Changing climate shifts timing of European floods. Science, 357(6351), 588-590. 2. Di Baldassarre, G., A. Montanari, H. Lins, D. Koutsoyiannis, L. Brandimarte, and G. Bloeschl (2010). Flood fatalities in Africa: from diagnosis to mitigation, Geophysical Research Letters, 37, L22402. 3. Jongman, B., Winsemius, H. C., Aerts, J. C., de Perez, E. C., van Aalst, M. K., Kron, W., & Ward, P. J. (2015). Declining vulnerability to river floods and the global benefits of adaptation. Proceedings of the National Academy of Sciences, 112(18), E2271-E2280. 4. Mechler, R., & Bouwer, L. M. (2015). Understanding trends and projections of disaster losses and climate change: is vulnerability the missing link? Climatic Change, 133(1), 23-35. 5. Wind, H. G., Nierop, T. M., de Blois, C. J., & de Kok, J. L. (1999). Analysis of flood damages from the 1993 and 1995 Meuse floods. Water Resources Research, 35(11), 3459-3465. 6. Di Baldassarre, G., Viglione, A., Carr, G., Kuil, L., Yan, K., Brandimarte, L., & Blöschl, G. (2015), Perspectives on socio-hydrology: Capturing feedbacks between physical and social processes, Water Resources Research, 51, 4770–4781. 7. Kreibich, H., Di Baldassarre, G., Vorogushyn, S., Aerts, J. C., Apel, H., et al. (2017). Adaptation to flood risk: Results of international paired flood event studies. Earth's Future, 5(10), 953-965. 8. White, G.F. (1945). Human Adjustments to Floods. Department of Geography. Research Paper No. 29. Department of Geography, University of Chicago, Chicago. 225 pages. 9. Kates, R. W., Colten, C. E., Laska, S., and Leatherman, S. P. (2006), Reconstruction of New Orleans after Hurricane Katrina: A research perspective. Proceedings of the National Academy of Sciences of USA, 103(40), 14653-14660. 10. Ludy J., & Kondolf, G. M. (2012).

Flood risk perception in lands "protected" by 100-year levees. Natural Hazards, 61(2), 829-842. 11. Di Baldassarre, G., Kemerink, J.S., Kooy, M., Brandimarte, L. (2014). Floods and societies: the spatial distribution of water-related disaster risk and its dynamics. Wiley Interdisciplinary Reviews: Water, 1(2), 133-139. 12. Mård, J., Di Baldassarre, G., Mazzoleni, M. (2018) Nighttime light data reveal how flood protection shapes human proximity to rivers. Science Advances, 4(8), eaar5779. 13. Auerswald, K., Moyle, P., Seibert, S. P., & Geist, J. (2019). HESS Opinions: Socio-economic and ecological trade-offs of flood management–benefits of a transdisciplinary approach. Hydrology and Earth System Sciences, 23(2), 1035-1044.

Please also note the supplement to this comment:
https://www.nat-hazards-earth-syst-sci-discuss.net/nhess-2018-336/nhess-2018-336-AC2-supplement.pdf
* * *
**NHESSD**
[Figure]

**Supplement:**

[revised manuscript text omitted]

---

## Author Comment (AC3) · 17 Apr 2019

Authors' responses to interactive comments on "Flood Risk in a Range of Spatial Perspectives – from Global to Local" by Z. W. Kundzewicz et al.

Anonymous Referee #1

This paper discussed a very important topic, the flood risk at global and regional scales. I appreciate the authors' effort to tackle this issue. But it requires a decent amount of work to solidify some statements.

Thanks for encouraging words. We tried to solidify problematic statements.

I suggest the authors extend and solidify the literature review in this paper. For example, the authors believe that the frequency of flood does not have obvious changes during the past. However, based on previous research, at least in the US, the lowfrequency floods do have a significant change, while the high-frequency floods do not have a significant change.

We did our best to extend the literature review, even if there are thousands of source items that could be referred to, so that we had to be selective. Indeed, we introduced additional references from the USA (Links and Slack, 1999, 2005; as well as Shaw & Riha, 2011) that solidify the review and demonstrate the subtleties of findings, demonstrating the lack of a one-size-fits-all result.

The characteristics of floods include are magnitude, frequency and timing. So I also suggest the authors extend a paragraph to discuss the influence of the change of floods' timing, instead of just one sentence in the current version. We devoted more room to the timing issues, referring to results by Blöschl et al. (2017) as well as Hodgkins et al. (2017). Please change the format of latitude and longitude. We changed the format by using superscripts.